# Effect of Acid-Etching Duration on the Adhesive Performance of Printed Polyetheretherketone to Veneering Resin

**DOI:** 10.3390/polym13203509

**Published:** 2021-10-13

**Authors:** Jiaqi Zhang, Yingjie Yi, Chenwei Wang, Ling Ding, Ruijin Wang, Guofeng Wu

**Affiliations:** 1Department of Prosthodontics, Nanjing Stomatological Hospital, Medical School of Nanjing University, Nanjing 210000, China; jdzhangjiaqi@163.com (J.Z.); yiyingjie0821@163.com (Y.Y.); wcw1751273982@126.com (C.W.); ding.lyn@yahoo.com (L.D.); hakusanbai@126.com (R.W.); 2Digital Engineering Center of Stomatology and Department of Prosthodontics, Nanjing Stomatological Hospital, Medical School of Nanjing University, Nanjing 210000, China

**Keywords:** 3D printing, polyetheretherketone (PEEK), sulfuric acid, adhesives, prosthodontics

## Abstract

Three-dimensional printing polyetheretherketone (PEEK) provides a new choice for dental prostheses, while its appropriate bonding procedure and adhesive performance are still unclear. This study aimed to investigate the adhesive performance of printed polyetheretherketone (PEEK) after acid etching to veneering resin. In total, 182 PEEK specimens (including 91 printed and 91 milled specimens) were distributed to 14 subgroups (*n* = 13/subgroup), according to the manufacturing process and surface treatment. The specimens were polished and etched with sulfuric acid for 0, 5, 30, 60, 90, 120, and 300 s, respectively. Two specimens in each subgroup were observed under a scanning electron microscope (SEM) for surface and cross-section morphology separately. Then, the specimens were treated with a bonding primer, and one specimen in each subgroup was prepared for cross-sectional observation under SEM. The residual 10 specimens of each subgroup bonded with veneering resin were tested with the shear bond strength tests (SBS) and failure modes analysis. Statistical analysis was performed by one-way ANOVA followed by the SNK-q post hoc test (*p* < 0.05). The etched pores on the PEEK surface were broadened and deepened under SEM over time. Printed PEEK etched for 30 s obtained the best SBS-to-veneering-resin ratio (27.90 ± 3.48 MPa) among the printed subgroups (*p* < 0.05) and had no statistical differences compared with milled PEEK etched for 30 s. The SBS of the milled subgroups etched from 5 to 120 s were over 29 MPa without significant between-group statistical differences. Hence, printed PEEK can be coarsened effectively by 30 s of sulfuric acid etching. The adhesion efficacy of printed PEEK to veneering resin was qualified for clinical requirements of polymer-based fixed dentures.

## 1. Introduction

Polyetheretherketone (PEEK), a high-performance polymeric material successfully utilized in traditional industrial fields, has been gradually introduced into dentistry and applied as a framework material of complete dentures, removable partial dentures, and implant-supported prostheses [1,2,3]. Compared with existing dental materials, PEEK shows better mechanical properties of flexural strength and fracture resistance [4,5]. Most reported PEEK prostheses were fabricated with the CAD/CAM milling technique, while the cost of excess material and time remain to be solved in the future [5,6,7]. Additive manufacturing, also named 3D printing, offers a remarkable utilization rate of the material and good shape ability, which has attracted the focus of researchers [8,9]. Currently, the most widespread 3D printing technologies include selective laser sintering (SLS), fused deposition modelling (FDM), stereolithography (SLA), digital light processing (DLP), laminated object manufacturing (LOM), and so forth. Fused deposition modeling (FDM) is one representative technique of 3D printing, by squeezing the filamentous fused thermoplastic material through a digitally controlled nozzle [10,11]. The thermoplastic materials (mostly polymers) were stacked layer by layer, ending up as the ultimate shapes. Alteration of printing direction or processing parameters would affect the mechanical properties of the final products [11].

Pigments including TiO_2_ powders were filled into PEEK to improve the unfavorable color of pure PEEK for clinical use, while this color-modified PEEK hardly improved the aesthetic demand [12]. Explorations achieved good effects by coating veneering resin onto the PEEK surface [5,13,14]. PEEK, as an inert material, has limited bonding strength compared to conventional adhesives. Before bonding to the veneering resin, appropriate surface treatment, including sandblasting, sulfuric acid etching, and bonding primer treatment, could be mechanically or chemically helpful to the definitive adhesive properties [15,16]. It has been demonstrated that milled PEEK etched by sulfuric acid (98%) for 60 to 120 s obtained reliable adhesive strength compared to resin composite [16,17]. Sulfuric acid (98%) could have sulfonic functionalization with PEEK, resulting in micro-scale valleys and pores over the PEEK surface, which could increase its surface roughness and wettability [18,19]. Moreover, acid-etching combined with bonding primer treatment improved its bonding performance with resin by the micro-interlocking of the bonding primer and the etched PEEK pores, and the studies on tensile and shear bond strength showed that the Visio.link was considered an appropriate bonding primer in the bonding of PEEK to veneering resin [20,21,22]. However, due to different processes, optimal etching duration and the microscopic details of the bonding interface are still unclear for the printed PEEK to veneering resin. This study aimed to investigate the bonding efficacy between the pretreated printed PEEK and veneering resin and observe the morphology of the bonding interface. The first null hypothesis was that etching duration would not influence the shear bond strength of veneering resin to PEEK. The second null hypothesis was that PEEK made by two manufacturing processes did not present any differences in terms of adhesive performance.

## 2. Materials and Methods

### 2.1. Specimen Preparation

#### 2.1.1. Fabrication of the Original Specimens

Table 1 shows the compositions and details of the materials used in this study. Ninety-one disk-shaped PEEK specimens (diameter of 10 mm and thickness of 2 mm) were milled from PEEK dental disks (Vestakeep, Evonik, Essen, Germany) by a milling machine (UF-2SS-V, Haas Automation, Oxnard, CA, USA) under the water coolant. Ninety-one printed PEEK specimens were prepared with PEEK filaments according to the same dimension. Table 2 summarizes the main printing parameters. Each specimen was polished with an automatic water-cooled polishing machine (Mecatech234, PRESI, Grenoble, France) using silicon carbide abrasive papers (Starcke; Matador, Germany) of 400-, 800-, 1200-, 1500-, 2000-, and 3000-grit for 6 min. Subsequently, all the specimens were thoroughly cleaned by an ultrasonic bath in distilled water for 15 min and dried with oil-free air (Figure 1). The experimental group was the printed group (P group), and the control group was the milled group (M group).

#### 2.1.2. Sulfuric Acid Etching

According to the etching duration of sulfuric acid, both the P group and M group were further divided into seven subgroups (*n* = 13/subgroup), respectively, which were etched with 98% sulfuric acid (Sinopharm Chemical Reagent, Shanghai, China) for 0, 5, 30, 60, 90, 120, and 300 s. Then, all specimens were ultrasonically cleaned in deionized water for 15 min to remove surface residues and dried with oil-free air. Two specimens in each subgroup were prepared for observation of surface morphology.

#### 2.1.3. Treatment with the Bonding Primer

The bonding primer (Visio.link, Bredent GmbH, Senden, German) was coated to the remaining 11 specimens in each subgroup with a micro brush and then blown with oil-free air for 20 s to form a thin layer. Subsequently, the bonding primer was polymerized for 90 s with an LED curing light (Bluephase Style, Ivoclar Vivadent AG, Schaan, Liechtenstein) in the wavelength range of 370–400 nm according to the manufacturer’s instruction. One specimen in each subgroup was selected randomly for the observation of cross-section morphology.

### 2.2. Microscopic Morphology and Surface Elemental Compositions

Two etched-only specimens in each subgroup were randomly selected and observed separately for surface and cross-section morphology using a scanning electron microscope (SEM) (S-4800, Hitachi, Tokyo, Japan) equipped with an energy-dispersive spectrometer (EDS). The specified specimens for cross-section observation were frozen in liquid nitrogen for 10 min and snapped under to obtain a clear cross-sectional view. Another specimen coated with the bonding primer in each subgroup was selected randomly for cross-sectional microscopic observation.

### 2.3. Shear Bond Strength Tests

The veneering resin (Ceramage, Shofu, Kyoto Prefecture, Japan) was shaped in a mold (4 mm in inner diameter, 5 mm in thickness), pressed onto the center of each PEEK specimen surface, and light-polymerized for 3 min. All the specimens were stored in distilled water at 37 °C for 24 h before testing. The specimens were mounted with a loading jig on a universal testing machine (CMT4204, MTS, Shanghai, China). The shear bond strength test was operated with a crosshead speed of 0.5 mm/min. The shear bond strength (MPa) was defined as the ratio between the load at failure (N) and the area of the bonded surface in square millimeters (mm^2^).

### 2.4. Failure Modes Analysis

The fractured surfaces of all tested specimens in 2.3.2. were observed under the stereomicroscope (SMZ1500, Nikon, Tokyo, Japan) at ×20 magnification. Failure modes were classified into four types: (1) adhesive failure between PEEK and resin materials, (2) cohesive failure within PEEK, (3) cohesive failure within resin materials, and (4) mixed failure with both cohesive and adhesive failures. Representative PEEK fractured surfaces were selected for further SEM observation at ×30 magnification.

### 2.5. Statistical Analysis

Statistical analysis was performed using one-way ANOVA, followed by the SNK-q post hoc test using the SPSS V26.0 (IBM, Armonk, NY, USA). The statistical significance was set at α = 0.05.

## 3. Results

### 3.1. Surface Elemental Compositions of the Etched PEEK

Figure 2 shows the energy dispersive spectrum of the etched PEEK surfaces. No statistical difference was found in the weight percent of sulfur element among the etched printed and milled PEEK subgroups.

### 3.2. Surface Morphology

The morphology of the etched surface (Figure 3) and cross-section (Figure 4 and Figure 5) under SEM showed that with the extension of etching duration, the etched pores were broadened. After the treatment of the primer, the cross-section SEM images of the etched interface at ×1000 and ×2500 magnification (Figure 4 and Figure 5) showed an increasing trend of the bonding primer penetrating depth along with the etching duration. The mean of etching thickness was given in Table 3.

### 3.3. Shear Bond Strength (SBS) Test

Among the printed PEEK subgroups, P30 presented the optimal SBS value 27.90 ± 3.48 MPa (*p* < 0.05). The SBS values of the M5, M30, M60, M90, and M120 subgroups were significantly greater than those of M0 and M300 (*p* < 0.05), while there was no statistical difference among the M5, M30, M60, M90, and M120 subgroups (Figure 6).

### 3.4. Failure Modes Analysis

Table 4 shows the counted failure modes. Figure 7 shows the relative morphology of fracture surfaces.

## 4. Discussion

The surface modification of PEEK is one of the major focuses prior to its extensive clinical application. In this present study, the bonding interface and the shear bond strength of acid-etched PEEK to veneering resin were systematically investigated. With the bonding-strength-to-veneering-resin ratio following the international standard (ISO 10477: 2020) (above 5 MPa) [23], the printed PEEK showed similar but not identical performance with milled PEEK. Thus, the null hypotheses of the study were rejected.

The sulfur element was detected from all the etched PEEK specimens’ surface, indicating the sulfonation reactions that happened on the PEEK surface. The sulfonation and the introduction of -SO_3_H groups on the PEEK surface have been proved in studies using Fourier Transform Infrared Spectroscopy (FTIR), thermal and chemical analyses, or energy dispersive spectrometry (EDS) [18,19,24,25,26]. The introduction of the -SO_3_H groups promoted the formation of the porous structure and enhanced PEEK wettability [25,27]. In addition to the sulfonation, PEEK was completely soluble in concentrated sulfuric acid at 25 °C, which is rarely mentioned in the previous studies’ focus on the surface modification of PEEK [28]. The solubility of PEEK in concentrated sulfuric acid is evidence against the prolongation of sulfuric acid etching duration. In addition, the SEM results showed that the dissolution of PEEK caused etching pores and destruction of the surface structure. The variation of etching pores and the deepening of the etching thickness were reported from the cross-sectional perspective for the first time. The etching pores and thickness of PEEK surfaces increased with the extension of the etching time resulting in weaker surface strength, especially in the groups of 120 and 300 s (Figure 5).

As shown in the adhesion interface from a pilot experiment (Figure 8), etched PEEK and veneering resin were stuck together by the primer and were not in direct touch with each other. In addition, Figure 4 shows that the primer penetrated part of the etched pores while the bottom part of the pores was still vacant. Thus, the adhesion interface included resin, primer, mixing layer (the etched PEEK penetrated by the primer, becoming thicker with the extension of etching duration), etched pores of PEEK, and the unetched PEEK. No mixing layer was observed from the unetched PEEK after primer treatment (Figure 4).

The shear bond strength (SBS) values showed that the adhesive properties of printed PEEK specimens were slightly lower than the milled groups to veneering resin. Among milled PEEK subgroups, the milled PEEK etched by concentrated sulfuric acid for 5 to 120 s (M5 to M120) achieved stable and reliable adhesive strength to veneering resin, consistent with conclusions from previous studies (60 to 120 s) [16]. Among the printed PEEK subgroups, the specimens etched for 30 s (P30) obtained the best SBS-to-veneering-resin ratio. Other experimental printed subgroups, though lower in SBS than P30 and corresponding milled PEEK subgroups, were still higher than P0, and above 5 MPa, the required value was ISO 10477:2020 [23]. Considering the SEM images, the prolonged etching time could bring wider etching pores for the primer to penetrate and to form a thicker mixing layer; however, it weakened the surface structure of PEEK, resulting in lower surface strength of PEEK. These results indicated that a longer etching duration might cause surface damage, and thus is not recommended by the authors. Considering the convenience of clinical practice and adhesive performance, the authors recommend the etching duration of 30 s as the optimal etching time for printed PEEK. For milled PEEK, in which group no statistical difference was observed from 5 s to 120 s, the duration of etching suggested by authors is less than 120 s.

In terms of failure modes analysis, no cohesive failure was observed in PEEK, and most failures occurred in the primer or resin layer among the milled PEEK subgroups. On the contrary, the stripping off of the etched surface layer of the printed PEEK was noticed (Figure 6), showing that the bonding strength between resin and the etched printed PEEK was greater than the interlayer bonding strength of the fused PEEK filaments. The weaker interlayer bonding strength in the printed PEEK may result in a weaker shear bond strength to the resin than that of milled PEEK. Furthermore, the printing parameters such as nozzle temperature, layer height, and wait-time were reported to have a profound influence on interlayer bonding strength in printed PEEK [29]. Future adjustments on the printing parameters to enhance the properties of printed PEEK are suggested by the authors.

One limitation of this study was the fact that no thermocycling or long-term water storage was carried out before the shear bond strength tests. According to most studies on adhesive performance, artificial aging procedures would have an impact on the shear bond strength, which was approximate to the actual bonding strength. Therefore, artificial aging procedures should be included to assess the stability of the bonding strength in subsequent studies. In addition, as a promising alternative material in dentistry, the biocompatibility of PEEK is still of significance. It was reported that the milled PMMA had less cytotoxicity than the traditional one, indicating that the manufacturing process could influence the residue of the ingredients and the biocompatibility of material [30]. Future studies on the biocompatibility of printed PEEK are also needed.

## 5. Conclusions

Based on the results of the study, the following conclusions were drawn:(1)The adhesive property of 3D-printed PEEK can satisfy the clinical needs of polymer-based fixed dentures according to ISO 10477:2020, although slightly lower than that of milled PEEK.(2)The appropriate etching duration of milled PEEK was less than 120 s since prolonged etching duration might cause surface damage and compromise the adhesive efficacy.(3)Thirty seconds was considered as the ideal etching duration for printed PEEK.(4)3D-printing procedures need to be improved for better interlayer bonding strength of PEEK and better surface adhesive performance.

## Figures and Tables

**Figure 1 polymers-13-03509-f001:**
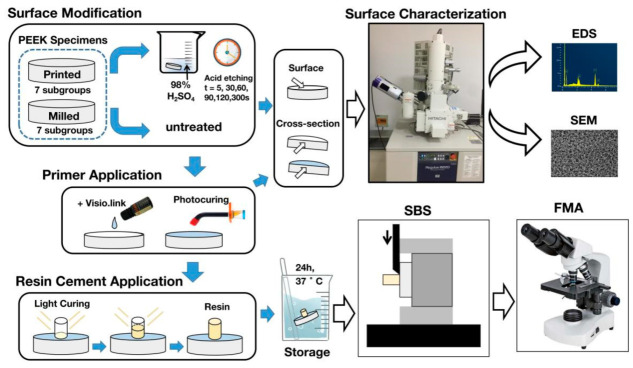
The diagram showing the workflow in this study. The surface modification indicates the pretreatment of PEEK. The surface characterization indicates microscopic morphology observation and surface elemental compositions analysis of PEEK. EDS, energy-dispersive spectrometer. SEM, scanning electron microscope. SBS, shear bond strength tests. FMA, failure modes analysis.

**Figure 2 polymers-13-03509-f002:**
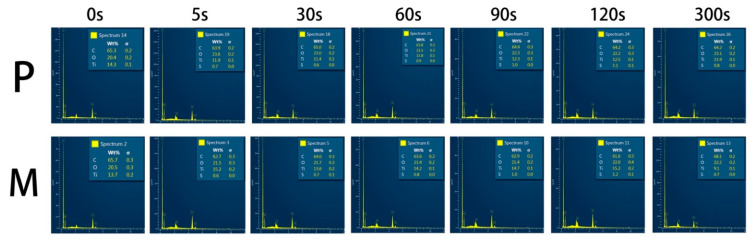
EDS spectrum of different sulfuric acid etching duration of the printed and milled PEEK specimens. **P**: etched printed PEEK; **M**: etched milled PEEK. EDS, energy-dispersive spectrometer.

**Figure 3 polymers-13-03509-f003:**
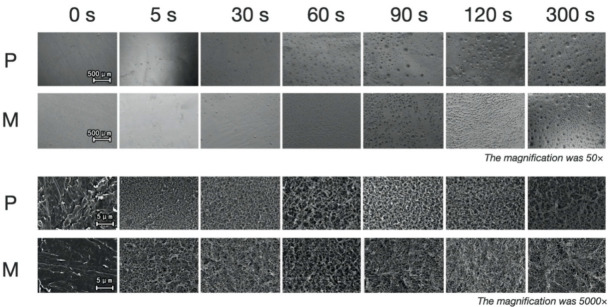
The SEM images showing the surface morphology of the etched PEEK at ×50 and ×5000 magnification. **P**: etched printed PEEK; **M**: etched milled PEEK.

**Figure 4 polymers-13-03509-f004:**
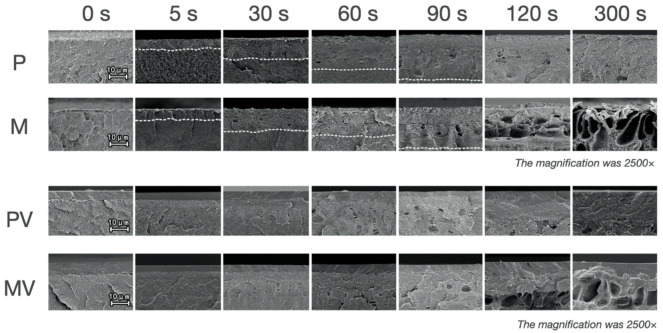
The SEM images showing the cross-section morphology of the etched PEEK before and after the treatment of the bonding primer at ×2500 magnification. **P**: etched printed PEEK; **M**: etched printed PEEK; **PV**: etched printed PEEK with the treatment of the primer Visio.link; and **MV**: etched milled PEEK with the treatment of the primer Visio.link. The dotted lines indicate the bottom of the etched pores. For overall views of the etched pores of the groups (etching duration 120 s and 300 s), the SEM images (×1000 magnification) were further supplied in Figure 5.

**Figure 5 polymers-13-03509-f005:**
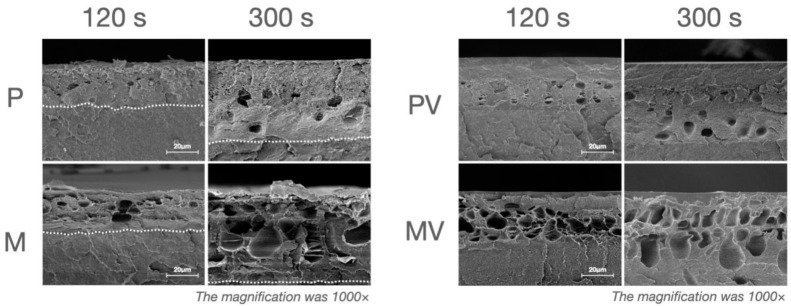
The SEM images showing the cross-section morphology of the etched PEEK before and after the treatment of the primer at ×1000 magnification. **P**: etched printed PEEK; **M**: etched printed PEEK; **PV**: etched printed PEEK with the treatment of the primer; and **MV**: etched milled PEEK with the treatment of the primer. The dotted lines indicate the bottom of the etched pores.

**Figure 6 polymers-13-03509-f006:**
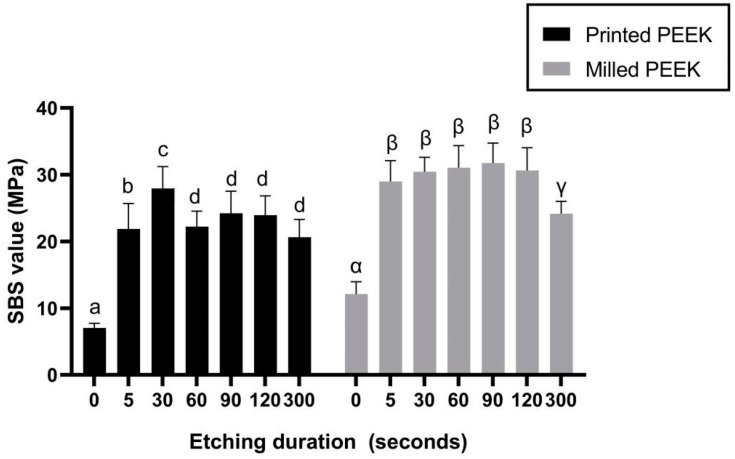
The mean ± SD of SBS values after different etching durations. Within the same color columns, different letters (a, b, c, α, β, and γ) indicate subgroups that are statistically different (*p* < 0.05). SBS, shear bond strength.

**Figure 7 polymers-13-03509-f007:**
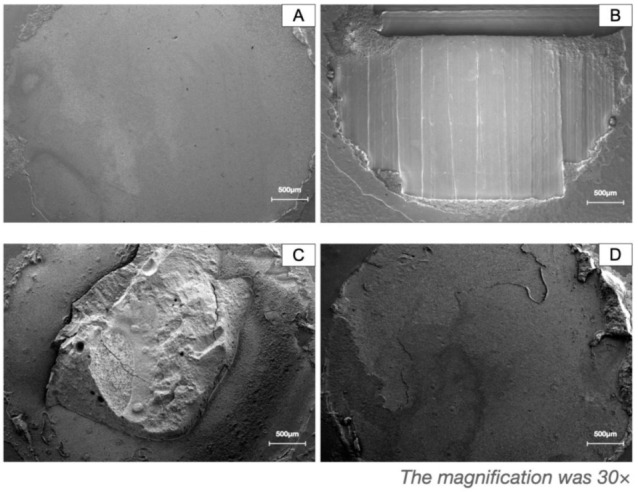
The representative failure modes of the bonding between PEEK and the veneering resin. (**A**) Adhesive failure between PEEK and resin materials, (**B**) cohesive failure within PEEK, (**C**) cohesive failure within resin materials, and (**D**) mixed failure with both cohesive and adhesive failures.

**Figure 8 polymers-13-03509-f008:**
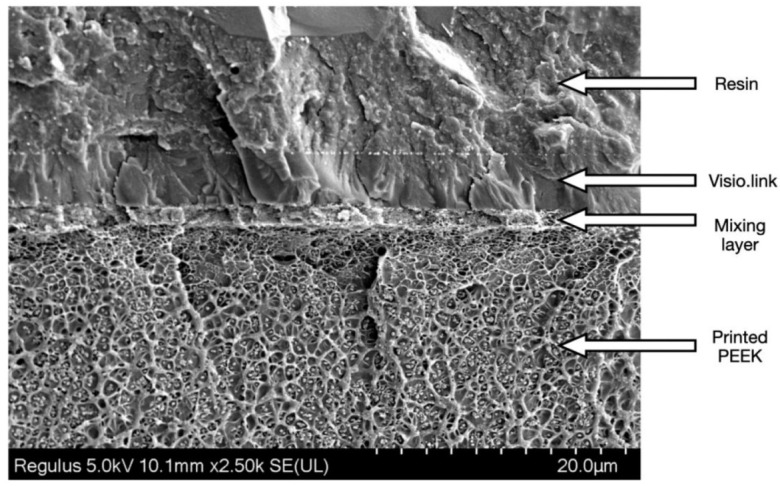
The typical bonding interface between the printed PEEK (etching for 5 s) and the veneering resin by the mediation of the primer Visio.link. Layers were labeled respectively.

**Table 1 polymers-13-03509-t001:** Main experimental material composition.

Materials		Main Composition	Manufacturers
PEEK compounds	PEEK disk	80% PEEK, 20% TiO_2_ pigments	Evonik, Germany
	PEEK filaments	80% PEEK, 20% TiO_2_ pigments	Evonik, Germany
Composite primer	Visio.link	MMA, Pentaerythritol triacrylate	Bredent, Germany
Light-curing veneer composite	Ceramage	Carbamate dimethacrylate, aluminum silicate glass, and hydroxyethyl methacrylate	Shofu, Japan

The main composition was based on the information provided by the manufacturers. PEEK, polyetheretherketone; MMA, methyl methacrylate.

**Table 2 polymers-13-03509-t002:** Parameters of the 3D printing process.

Description	Value
Filament diameter	1.75 mm
Nozzle temperature	410 °C
Nozzle diameter	0.4 mm
Heated building chamber	180 °C
Layer thickness	0.2 mm
Raster angle	Consistent with the longest edge
Printing speed	20 mm/s
Slicing software	Medvance Vulcan v2.1

**Table 3 polymers-13-03509-t003:** Sulfuric acid etching time points applied to the different groups.

Etching Duration (Seconds)	Etching Thickness (μm)	
	Printed PEEK	Milled PEEK
0	0	0
5	5.61 ± 0.17	6.18 ± 0.58
30	11.76 ± 0.21	11.84 ± 0.45
60	19.58 ± 0.23	18.08 ± 0.54
90	26.86 ± 0.28	24.05 ± 0.29
120	31.38 ± 1.39	29.60 ± 1.15
300	58.92 ± 0.88	62.86 ± 1.38

Data are Mean ± SD based on five measurements of etching thickness shown in Figure 4 and Figure 5 before the application of primer.

**Table 4 polymers-13-03509-t004:** Failure modes of each testing group after different surface treatments.

Etching Duration (Seconds)	Printed PEEK				Milled PEEK			
	A	AC	C-P	C-r	A	AC	C-P	C-r
0	10	0	0	0	9	1	0	0
5	7	0	3	0	8	2	0	0
30	5	4	1	0	4	5	0	1
60	7	1	2	0	3	6	0	1
90	6	1	3	0	3	5	0	2
120	5	1	3	0	5	4	0	1
300	6	2	2	0	5	5	0	0

A, adhesive failure; AC, combination of cohesive and adhesive failure; C-P, cohesive failure of PEEK; and C-r, cohesive failure of veneering resin.

## Data Availability

The data presented in this study are available on request from the corresponding author.

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
