# Peer review of "Effect of Acid-Etching Duration on the Adhesive Performance of Printed Polyetheretherketone to Veneering Resin"

_polymers, 2021, doi:10.3390/polym13203509_

Round 1

Reviewer 1 Report

This is a very interesting work on the adhesive ability of PEEK samples treated with different acid etching times followed by the application of a resinous adhesive.
The work is certainly of scientific interest but presents some criticisms listed below:

  • An opening sentence in the abstract section on the general problem must be inserted
    -At the end of the abstract section indicate which clinical requirements and consequences can be hypothesized
    -Check that all the keywords are Mesh terms of the Pubmed database
    -Line 40 before talking about FDM it is necessary to list, even if only briefly, the 4 most widespread techniques of additive 3D printing
    -Insert at the end of the introduciton section the null hypotheses of the study that must be refuted in the light of the results obtained in the study
    -explain why this geometry of the PEEK samples was chosen
    - table 3 must be removed because it is useless; just write the specifications in the text
    -line 106 the commercial name of the UV lamp is missing
    - subsections 2.3 are too many and cause confusion in the reader; I recommend bringing them together in a single, more discursive paragraph
    -SBS value tends to decrease in all samples at treatments of 300 sec; can the authors justify these results?
    -Very interesting figure 8 of which I congratulate the authors
    -In the discussion section I would have expected some more considerations on the choice of 30 sec as the ideal etching time for PEEK. In particular, no mention has been made of the possibility that a longer time may cause surface damage to the samples such as to compromise their adhesive performance and the need to use other techniques, such as AFM or 3D microscopy for the evaluation of these characteristics.
    -Another aspect to discuss in this section, in my opinion, is that of the biocompatibility of the peek since many studies in the literature are focusing on this aspect. In this regard, I recommend that you insert the following scientific work in the reference section, which could be of help to the reader:

  • Pagano S, Lombardo G, Caponi S, Costanzi E, Di Michele A, Bruscoli S, Xhimitiku I, Coniglio M, Valenti C, Mattarelli M, Rossi G, Cianetti S, Marinucci L. Bio-mechanical characterization of a CAD / CAM PMMA resin for digital removable prostheses. Dent Mater. 2021 Mar; 37 (3): e118-e130. doi: 10.1016 / j.dental.2020.11.003. Epub 2020 Nov 27. PMID: 33257084.

Reviewer 2 Report

the study could be interesting in the dental field, and was conducted with correct methodology. I have only a little doubts about the conclusions. Before proceeding to accept the manuscript I believe they need to be addressed

  1. describe in detail the limitations of this study
  2. provide more exhaustive descriptions to figure 1 and tables
  3. Formulate the null hypothesis of the study more clearly
  4. From what I was able to understand from your study and that in reality the etching time exceeded 5 seconds does not affect statistically significantly, at least to justify an increase in the exposure time to sulfuric acid, moreover the major differences I believe are more between the Printed and milled group. (if I'm wrong, correct me). If it is possible, I would add some statistical analyzes that reinforce the following statement: Printed PEEK etched by sulfuric acid (98%) for 30 seconds obtained the optimal shear bond strength to veneering resin. The appropriate etching duration of milled PEEK was less than 120 seconds.

Round 2

Reviewer 2 Report

The authors have answered all the questions posed in the first round of revision, They have modified the manuscript where requested, I consider the manuscript in the present form worthy of publication

Author Response

Thank you very much for your time and kind remarks. The authors deeply appreciate the valuable comments and help from the reviewers.